# Evaluation of *ex vivo* drug combination optimization platform in recurrent high grade astrocytic glioma: An interventional, non-randomized, open-label trial protocol

**Tan Boon Toh**[1,2]*, **Dexter Kai Hao Thng**[1,3], **Nagarjun Bolem**[4], **Balamurugan A. Vellayappan**[5], **Bryce Wei Quan Tan**[3], **Yating Shen**[1,3], **Sou Yen Soon**[6], **Yvonne Li En Ang**[6], **Nivedh Dinesh**[4], **Kejia Teo**[4], **Vincent Diong Weng Nga**[4], **Shiong Wen Low**[7], **Pek Lan Khong**[8,9], **Edward Kai-Hua Chow**[1,2,3], **Dean Ho**[1,2], **Tseng Tsai Yeo**[4], **Andrea Li Ann Wong**[3,6]*

**1** The N.1 Institute for Health (N.1), National University of Singapore, Singapore, Singapore, **2** The Institute for Digital Medicine (WisDM), National University of Singapore, Singapore, Singapore, **3** Cancer Science Institute of Singapore (CSI), National University of Singapore, Singapore, Singapore, **4** Division of Neurosurgery, Department of Surgery, National University Hospital, Singapore, Singapore, **5** Department of Radiation Oncology, National University Cancer Institute, Singapore, Singapore, **6** Department of Haematology-Oncology, National University Hospital, Singapore, Singapore, **7** Division of Neurological Surgery, Ng Teng Fong General Hospital, Singapore, Singapore, **8** Department of Diagnostic Imaging, National University Hospital, Singapore, Singapore, **9** Clinical Imaging Research Centre (CIRC), National University of Singapore, Singapore, Singapore

* lsittb@nus.edu.sg (TBT); Andrea_LA_WONG@nuhs.edu.sg (ALAW)

## Abstract

### Introduction

High grade astrocytic glioma (HGG) is a lethal solid malignancy with high recurrence rates and limited survival. While several cytotoxic agents have demonstrated efficacy against HGG, drug sensitivity testing platforms to aid in therapy selection are lacking. Patient-derived organoids (PDOs) have been shown to faithfully preserve the biological characteristics of several cancer types including HGG, and coupled with the experimental-analytical hybrid platform Quadratic Phenotypic Optimization Platform (QPOP) which evaluates therapeutic sensitivity at a patient-specific level, may aid as a tool for personalized medical decisions to improve treatment outcomes for HGG patients.

### Methods

This is an interventional, non-randomized, open-label study, which aims to enroll 10 patients who will receive QPOP-guided chemotherapy at the time of first HGG recurrence following progression on standard first-line therapy. At the initial presentation of HGG, tumor will be harvested for primary PDO generation during the first biopsy/surgery. At the point of tumor recurrence, patients will be enrolled onto the main study to receive systemic therapy as second-line treatment. Subjects who undergo surgery at the time of recurrence will have a second harvest of tissue for PDO generation. Established PDOs will be subject to QPOP analyses to determine their therapeutic sensitivities to specific panels of drugs. A QPOP-

**Data Availability Statement:** No datasets were generated or analysed during the current study. All relevant data from this study will be made available

in local and international journals upon study completion.

**Funding:** ALAW received funding from the National University Health System (NUHS) Seed Fund (NUHSRO/2021/052/RO5+6/Seed-Mar/01) and TTB obtained funding from the National University of Singapore Institute for Digital Medicine (WisDM) Program Seed Fund (WisDM/Seed/002/2021). The funders had no role in study design, data collection and analysis, decision to publish, or preparation of the manuscript.

**Competing interests:** I have read the journal's policy and the authors of this manuscript have the following competing interests: EKHC and DH are shareholders in KYAN Technologies. This does not alter our adherence to PLOS ONE policies on sharing data and materials.

**Abbreviations:** 2D, 2-dimensional; ALT, Alanine transaminase; ANC, Absolute neutrophil count; AST, Aspartate transaminase; BBB, Blood-brain barrier; DCE-MRI, Dynamic Contrast Enhanced-Magnetic Resonance Imaging; ECOG, Eastern Cooperative Oncology Group; FBC, Full blood count; Ga68-NEB, Gallium-68 NOTA-Evans Blue; HGG, High grade astrocytic glioma; MRI, Magnetic Resonance Imaging; NCI-CTCAE, National Cancer Institute Common Terminology Criteria for Adverse Events; OS12, Twelve-month overall survival; PDO, Patient-derived organoids; PET, Positron Emission Tomography; PFS6, Six-month progression-free survival rate; PT, Prothrombin time; QPOP, Quadratic Phenotypic Optimization Platform; RANO, Response Assessment in Neuro-Oncology; RT, Radiotherapy; TMZ, Temozolomide; ULN, Upper limit of normal.

guided treatment selection algorithm will then be used to select the most appropriate drug combination. The primary endpoint of the study is six-month progression-free survival. The secondary endpoints include twelve-month overall survival, RANO criteria and toxicities. In our radiological biomarker sub-study, we plan to evaluate novel radiopharmaceutical-based neuroimaging in determining blood-brain barrier permeability and to assess *in vivo* drug effects on tumor vasculature over time.

## Trial registration

This trial was registered on 8<sup>th</sup> September 2022 with ClinicalTrials.gov Identifier: NCT05532397.

## Background

High grade astrocytic glioma (HGG) is one of the most devastating human adult solid malignancies with limited therapeutic options. Despite advanced surgical intervention with adjuvant radiation and chemotherapy with temozolomide (TMZ/RT), HGG patients typically have a mean survival period of 12–15 months following diagnosis [1]. In addition, HGG tumors demonstrate vast intra- and inter-tumor heterogeneity that confer treatment resistance which eventually result in tumor recurrence in more than 90% of all HGG patients [2–4]. Currently, there is no standard treatment for recurrent HGG and the option of repeat surgical intervention is limited to about 25% of all recurrent patients [4]. Clearly, there is an urgent need to develop better systemic therapy strategies to improve patient outcome.

One of the major obstacles to therapeutic development is the lack of clinically relevant models. At present, U87MG, U251, and T98G are some of the most commonly used two-dimensional (2D) cell research models due to their ease of access and growth. However, 2D cell cultures present limitations due to the lack of tumor heterogeneity and cell-cell/matrix interactions that are present in primary patient tumor tissues [5]. While the National Cancer Institute 60 Panel (NCI-60) was established in an attempt to represent the heterogeneity observed across multiple cancer types, these 2D cell cultures still lack the comprehensive diversity exhibited in the clinics, and is thus being phased out currently [6]. Comparative studies conducted between the molecular profiles of cell lines and patient tumors have reported that conventional cell lines are only able to capture a subset of cancer types, and unable to represent the full array of cancer subtypes [7]. Recently, patient-derived organoids (PDOs) have been shown to faithfully recapitulate the heterogeneity observed in several primary cancer types including HGG, and are being utilized as an *in vitro* platform to evaluate patient drug sensitivity and the genomic profiles associated with each cancer type [8–12]. In the context of HGG specifically, organoids derived from patient-derived material faithfully recapitulated the heterogeneous cell identity observed in parental tumours [8, 9]. This was evidenced in the maintenance of the heterogeneous expression of stem cell markers such as SOX2, OLIG2 and NESTIN [8, 9]. Furthermore, the HGG PDOs were shown to retain the mutational profiles of parental tumours, including genetic aberrations frequently observed in gliomas such as *PTEN* loss and the gain-of-function *EGFR* variant III (*EGFRvIII*) [9]. Notably, by preserving the molecular profiles of the parental tumors, PDOs serve as an effective tool for the investigation of drug responses and mechanisms *ex vivo* [9–12]. Drug sensitivity studies in HGG PDOs have exhibited strong correlation between mutational status of a routinely performed clinical sequencing gene panel, and the sensitivities of the HGG PDOs to its corresponding targeted therapies as well as

immunotherapy [9]. For instance, HGG PDOs harbouring EGFR aberrations exhibited preferential sensitivity to first-generation EGFR inhibitor, gefitinib, while the presence of secondary mutations rendered the PDOs resistant [9]. Human PDOs therefore represent more clinically relevant models compared to traditional cell line-based models, with the ability to recapitulate disease heterogeneity for drug response studies with significant clinical relevance.

Identifying the optimal patient-specific drug combinations, however, remains a challenge. This is attributed to the complex molecular networks that drives feedback mechanisms giving rise to drug resistance and compensatory activation of oncogenic drivers, thus limiting the efficacy of current targeted therapies. Current biomarker-based approaches in predicting drug sensitivity and clinical outcome are limited only to select patient subpopulations and specific drugs and are still not broadly applicable [13]. These approaches are largely limited to monotherapies and are unable to identify patient-specific drug combinations. Furthermore, biomarker-driven clinical management of HGG are broadly used for targeted therapies such as EGFR inhibitors, with few biomarkers predicting chemosensitivity of HGG patients [14, 15]. Developing and evaluating therapeutic platforms that can provide clinicians with additional therapeutic sensitivity information at the patient-specific level may therefore improve therapeutic outcomes in HGG patients, particularly in the setting of tumor recurrence, where there are several chemotherapeutic options but no established standard of care [16–20]. As an experimental-analytical hybrid drug combination interrogation platform, QPOP can model the expected patient response to panel of drugs by systematically perturbing biological model systems such as patient-derived cells or organoids to the drugs in defined combinations derived from an orthogonal-array composite design [21]. The functional responses of the patient-derived avatars are then used to establish a second-order regression model from which all possible combinations of the drug panel can be ranked [21]. QPOP has been successfully applied in patients with refractory hematological malignancies such as multiple myeloma and lymphomas, yielding favourable clinical response in patients [21–24].

This proposal therefore aims to test the feasibility of QPOP application in HGG, by first establishing clinically relevant HGG organoid models, followed by the utility of QPOP-derived drug combinations in HGG. We hypothesize that optimal QPOP-derived drug combinations from clinically relevant HGG organoids can direct management and improve outcomes of HGG patients. In addition, given the nature of the blood-brain barrier (BBB) which poses significant challenges to the delivery of systemic therapy to brain tumors, this study presents a unique opportunity to examine the relationship between individualized tumoral therapeutic sensitivity assessed by novel radiopharmaceutical-based neuroimaging and clinical outcomes. Overall, our study seeks to justify the application of QPOP for personalized cancer therapy in HGG patients by identifying and clinically validating QPOP-guided drug combinations in the recurrent setting.

## Methods/Design

### Objectives

This study seeks to achieve three specific aims.

**Specific Aim 1**: To establish primary and corresponding TMZ/RT-resistant HGG PDOs. Our central hypothesis is that PDOs mimic the biological characteristics of HGGs and serve as an ideal platform for the evaluation of drug sensitivities, accurately reflecting the patient's therapeutic response to the drugs. By further comparing the sensitivity profiles of TMZ/RT-resistant organoids with those of samples obtained during the second surgery at the

time of recurrence, we will be able to determine how closely laboratory-generated resistance mimics natural treatment-induced resistance.

**Specific Aim 2**: To determine the utility of QPOP-derived drug combinations in treating recurrent HGG by detecting signals of efficacy in identifying patient-specific drug combinations using an n-of-1 approach, which will then need to be confirmed in a larger cohort Phase II study. In this study, we will specifically evaluate patient specific top-ranking combinations in patients with recurrent HGG, with the objective of determining if QPOP analysis can effectively predict therapeutic sensitivities as well as determine the optimal treatment regime with improved treatment outcomes compared to historical controls at the time of relapse. HGG is an ideal tumor type to test our hypothesis because there is currently no standard-of-care second line systemic therapy for this disease.

**Specific Aim 3**: In a radiological biomarker substudy, we aim to evaluate how BBB permeability evolves over time from first diagnosis to tumor recurrence using Gallium-68 NOTA-E-vans Blue (Ga68-NEB) Positron Emission Tomography/ Magnetic Resonance Imaging (PET/MRI) and Dynamic Contrast Enhanced (DCE)-MRI imaging, subsequently correlating imaging biomarkers with pathological biomarkers, QPOP-derived therapeutic response, clinical and standard radiological outcomes.

## Trial design

This is an interventional, non-randomized, open-label single site study (Fig 1). The primary purpose of this study is not hypothesis testing, but to assess the feasibility of QPOP-guided therapy for recurrent GBM to be used in a larger scale study. Therefore, our study does not have a formal sample size, but rather, a set benchmark to determine feasibility. We plan to

|  | Pre-screening phase | | | | Main study - at point of relapse | | | | |
|  | Before surgery | During surgery | Before intervention | Intervention | Pre-treatment | At the start of each cycle thereafter | | | Close-out |
| **TIMEPOINT** |  |  |  |  | $t_{-2\ weeks}$ | $t_0$ | $t_{3\ days}$ | $t_{8\ weeks}$ | Final visit [a] |
| **ENROLMENT:** |  |  |  |  |  |  |  |  |  |
| Informed consent | X |  |  |  | X | X |  |  |  |
| Medical history |  |  |  |  | X | X |  |  | X |
| *Physical examination* |  |  |  |  | X | X |  |  | X |
| *Concomitant medication notation* |  |  |  |  | X | X |  |  | X |
| **INTERVENTIONS:** |  |  |  |  |  |  |  |  |  |
| *Stupp protocol* |  |  |  | ←—→ |  |  |  |  |  |
| *QPOP-guided combinations* |  |  |  |  |  | ←————————→ | | |  |
| **ASSESSMENTS:** |  |  |  |  |  |  |  |  |  |
| *ECOG performance status* |  |  |  |  | X | X |  |  | X |
| *Toxicity assessments (CTCAE v 5.0)* |  |  |  |  |  | X |  |  | X |
| *Haematology (Full blood count)* |  |  |  |  | X | X | X |  |  |
| *Blood chemistry* |  |  |  |  | X | X | X |  |  |
| *Radiologic tumour measurement* |  | X |  |  | X | X |  | X |  |
| *Blood Sampling for research purposes* |  | X |  |  | X | X |  | X |  |
| *Tumor sample collection [b]* |  | X |  |  | X [c] |  |  |  |  |

**Fig 1. Schedule of enrolment, interventions and assessments.** [a] Final study visit procedures can be done within 30 days of study discontinuation/completion of study treatment. [b] If surgery is deemed to be too risky for patient at pre-screening phase, patient biopsy sample will be used instead if there is sufficient tumor tissue. In addition, tumor may be harvested for any subsequent procedure that patient might require for brain tumor treatment. [c] If there is a second tissue collection at relapse.

enroll a total of 10 patients who receive QPOP-guided chemotherapy at the time of first high grade astrocytic glioma recurrence over two years from 17 February 2023 to 31 December 2025. We expect this to be sufficient for determining the feasibility of a phase II trial in which there will be formal sample size calculations for the achievement of six-month progression-free survival (PFS6) of at least 30% [25]. In addition, a phase II study will only be considered feasible if there is successful organoid generation and QPOP analysis in > 50% of all tumour samples collected at the first resection. Successful QPOP analysis will be defined by a $z' > 0.5$ for the test drug combination drug screening assay and $R^2 > 0.7$ for QPOP analysis of experimental data. In this study, subjects will be replaced accordingly if their QPOP analysis is unsuccessful or its results do not lead to therapeutic decision making.

**Pre-screening phase.**   Subjects will be approached for pre-screening consent at the time of first suspected diagnosis of a HGG and tumor will be harvested at the time of initial surgery/biopsy for the generation of PDOs. Once the histological diagnosis of HGG is confirmed and the subject is planned for adjuvant TMZ/RT [26], our pre-screening cohort will undergo baseline standard gadolinium-enhanced MRI plus investigational Ga68-NEB PET/MRI and DCE-MRI imaging within one week of the planned radiotherapy simulation date. Subjects who do not fulfil the histological criteria or will not be receiving standard adjuvant temozolomide/ radiotherapy will be considered screen failures and will not be included in the main study. It is estimated that we will need to pre-screen approximately 20 subjects in order to achieve our objective of enrolling 10 recurrent high grade glioma subjects treated with QPOP-guided systemic therapy. Subjects will then undergo standard-of-care treatment for HGG with concurrent TMZ/RT, with clinical examination, laboratory tests and MRI imaging performed at regular intervals, as per institutional guidelines for standard clinical care. Upon completion of adjuvant TMZ/RT, subjects will be monitored for signs and symptoms of relapse, as well as radiological progression.

HGG organoids will be generated from primary tumor samples and QPOP analyses of the treatment-naïve HGG organoids will be performed (QPOP-1) (Fig 2). To model TMZ/RT-resistant cells in the setting of recurrent HGG, these treatment-naïve organoids will be subjected to chronic exposure of TMZ, and ionizing radiation. QPOP analyses will be performed on these resistant HGG organoids (QPOP-2) and the results compared with those of QPOP1 (Fig 2).

**Main study.**   At the time of documented tumor recurrence, eligible subjects will be reassessed for suitability to participate in the main study and approached for their informed consent to enter this phase of the study. At this juncture, a small proportion of subjects will be deemed suitable for a second operation. If this is the case, HGG organoids will also be generated from the recurrent HGG tumor and subject to QPOP analyses (QPOP-3) (Fig 2). During the main study, subjects will receive QPOP-guided systemic therapy for the treatment of their relapsed HGG and will be assessed regularly for safety and efficacy of this therapy. MRI brain as well as investigational Ga68-NEB PET/MRI and DCE-MRI imaging will be performed prior to and at 8 weeks (+/- 1 week) after initiation of systemic therapy. Beyond that, MRI brain will be used to reassess disease status every 8 weeks (+/- 1 week) while on QPOP-guided systemic therapy until unequivocal disease progression.

## Dosing regimen

We will test our PDOs against two *in-vitro* drug screening panels. The "clinical care panel" (Panel 1) comprises anti-cancer agents which have previously been evaluated in recurrent HGG: SN-38, lomustine, temozolomide, vincristine, procarbazine, etoposide, carboplatin, cisplatin, cyclophosphamide, docetaxel, capecitabine, vinorelbine [16–18, 26–35]. All of these drugs have been determined to have sufficient penetration across the blood-brain barrier and

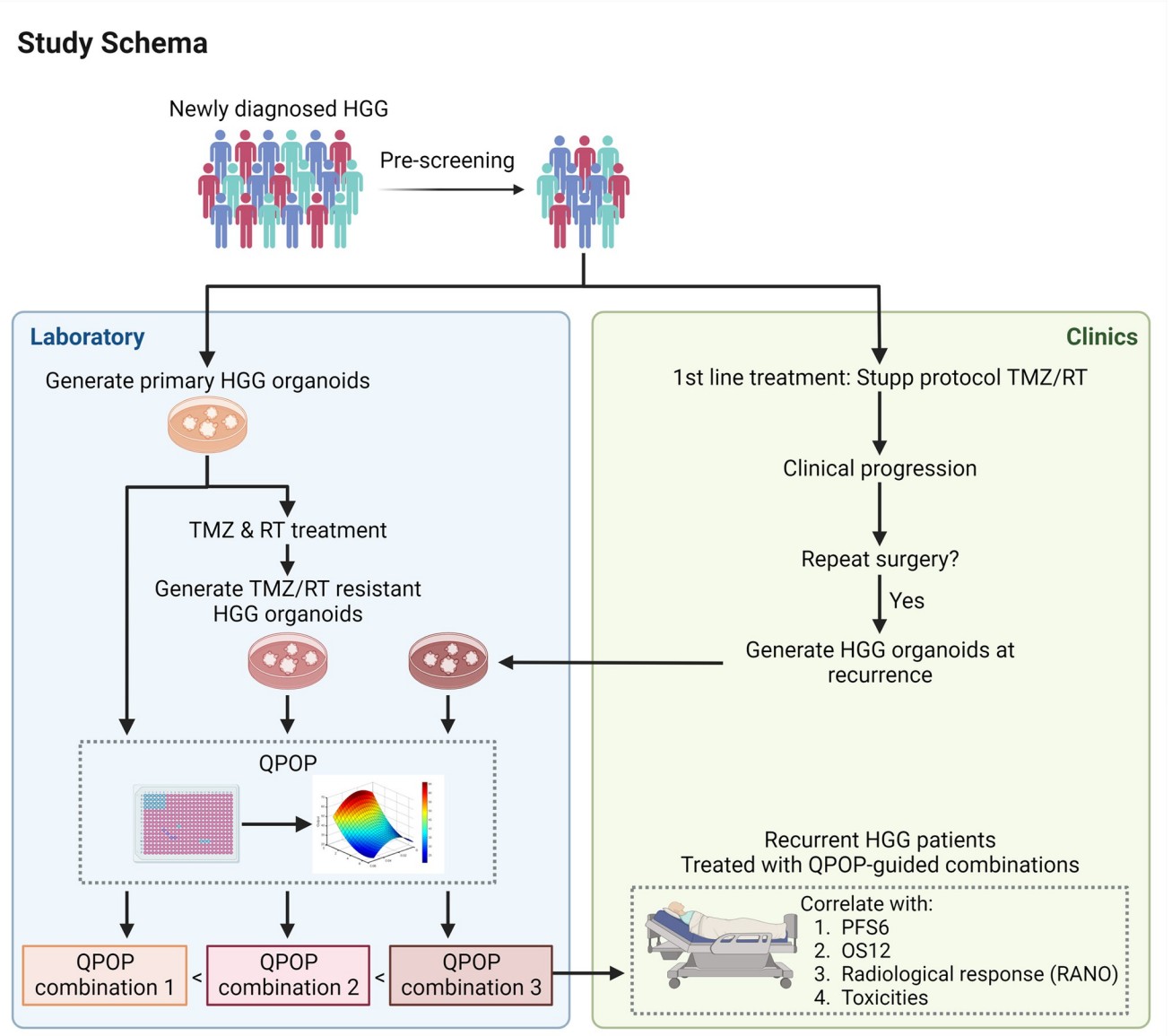

**Fig 2. General study schema for QPOP-guided drug combination therapy.** Subjects will be enrolled to the pre-screening phase of our study at the point of first diagnosis of HGG. Established HGG PDOs from newly diagnosed subjects will be subjected to continuous low dose TMZ and RT as described by the Stupp protocol to model the treatment received in the clinics. QPOP results will be generated from primary HGG PDOs (**QPOP-1**), matched TMZ/RT-resistant HGG PDOs (**QPOP-2**). At the point of recurrence, subject will be approached for consent for the main protocol if the PDO generation from their primary tumors was successful. If the subject is found to be suitable for surgery at the time of recurrence, tissue will be collected for the generation of PDOs at recurrence (**QPOP-3**). QPOP-guided second line systemic treatment will be used in order of preference with available QPOP data from QPOP-3> QPOP-2> QPOP-1. Figure was created with BioRender.com.

have known adverse event profiles. QPOP will be used to derive top-ranking drug combinations, which will then be validated *in vitro*. The "experimental panel" (Panel 2) comprises anticancer agents which have promising activity in HGG but are not yet routinely used in HGG patients: Geftinib, osimertinib, vemurafenib, everolimus, sunitinib, regorafenib, selinexor, marizomib, abemaciclib, ivosidenib, olaparib, metformin. The results of this panel will not be used in the care of our enrolled subjects; instead, they are hypothesis-generating and intended for us in designing future trials.

The drug combinations generated by QPOP can be divided into 3 categories: 1) combination regimens with published data in the setting of gliomas (category 1); 2) combination regimens where there is published data on intracranial activity and anti-glioma effect of the individual agents (either as monotherapy or in combination with other agents), and where there is published safety data on the combination in non-glioma settings (category 2); and 3) novel drug combinations that have not been assessed for clinical efficacy (category 3). We will apply the following treatment selection algorithm in determining the drug regimen of choice for individual subjects:

- If QPOP analysis results of combinations 1, 2 and 3 are all available at the time the subject is due to start systemic therapy for the treatment of recurrent HGG (i.e., 4–6 weeks after surgery), they will be used in the following order of preference: QPOP-3> QPOP-2> QPOP-1, provided other eligibility criteria for the use of that particular combination's results are met (Fig 2).

- We will preferentially use category 1 combinations, but will consider category 2 combinations should the QPOP analysis not generate any category 1 combinations. Decisions to choose category 2 combinations vs defaulting to monotherapy (see point 3) will be discussed with the study team. Combination therapy would be prioritised over monotherapy and monotherapy will only be applied when QPOP combination therapy results do not fall within the criteria for usage as category 1 or category 2 combinations. Doses of drug combinations used should be as close as possible to those used in the published literature. We will not use novel drug combinations (category 3) that have not previously been assessed for clinical safety.

- If the QPOP combination therapy results do not fall within the criteria for usage as category 1 or category 2 combinations, monotherapy may be considered. Temozolomide monotherapy will not be encouraged unless strongly supported by QPOP-2 or -3 results, in which case, a dose schedule different from first-line therapy should be used.

- Bevacizumab is an FDA-approved drug for the treatment of recurrent HGG (Avastin (bevacizumab) injection, Genetech, Inc, December 2017), and may be used in combination with lomustine [16], temozolomide [16, 18], irinotecan [32], carboplatin [31] and etoposide [31] for the treatment of HGG. However, bevacizumab is not included in the drug screening panel because our current PDO models lacks vasculature, hence the functional testing of anti-angiogenic drugs is irrelevant. In order to be relevant to real-world practice, the study will allow the addition of bevacizumab to any of the above agents, provided that investigators adhere as closely as possible to the regimens in the referenced studies. The addition of bevacizumab will not be permitted if the QPOP results suggest an effective combination where there is no published data for the addition of bevacizumab to that particular treatment combination. Deviations from existing data may be permitted only with the agreement of the study team.

The study team will meet to review the QPOP analysis results to agree on the validity of the data and determine the recommended treatment at the time of relapse for each subject on the study. As an additional safety measure, the study team will also conduct a combined review for each subject at the time of their first post-treatment radiologic assessment, to ensure the appropriateness of therapy based on the efficacy and toxicity of the treatment selected.

## Efficacy endpoints

The primary clinical endpoint is six-month progression-free survival (PFS6). PFS is defined as the time from the start of study treatment to documented progression of disease or death;

PFS6 refers to the proportion of subjects who are alive and free of HGG progression at 6 months.

The three secondary clinical endpoints are:

1. Radiological response assessments at follow up MRI. The determination of radiographic response is as per the Response Assessment in Neuro-Oncology (RANO) criteria [36].

2. Twelve-month overall survival (OS12). OS is defined as the length of time from the start of study treatment, that subjects diagnosed with the disease are still alive. OS12 refers to the proportion of subjects who are alive at 12 months.

3. Hematological and non-hematological toxicities. As defined by National Cancer Institute Common Terminology Criteria for Adverse Events (NCI CTCAE) version 5.0

## Subject population

Male and female subjects aged 21 years and above with suspected high grade astrocytic glioma planned for surgery or biopsy followed by adjuvant chemoradiotherapy will be invited to participate in the pre-screening study. Subjects will only be enrolled in the main study if they had pathologically confirmed HGG, and received adjuvant treatment comprising standard-of-care therapy with surgery/biopsy followed by temozolomide and radiotherapy. In addition, they are required to have had sufficient tumor tissue available for PDO generation at baseline. This study is expected to take 18–24 months to complete. Written informed consent for entry into the study will be obtained prior to any study specific procedure. All eligibility criteria and consent forms will be checked before treatment is initiated.

**Inclusion criteria.** The following inclusion criteria apply only to the pre-screening phase:

1. Subjects 21 years of age or older, with Eastern Cooperative Oncology Group (ECOG) performance status 0 to 2, and with a life expectancy of more than 3 months with suspected high grade astrocytic glioma, fit for treatment comprising standard-of-care therapy with adjuvant temozolomide and radiotherapy if the diagnosis of high grade astrocytoma is pathologically confirmed.

2. Signed informed consent obtained before any study specific procedure. Subjects must be able to understand and be willing to sign the written informed consent.

 * Subjects will be enrolled at the time of initial surgery but study imaging and further PDO generation will not take place if the subject is subsequently found not to meet the histological criteria or will not be receiving standard adjuvant temozolomide/ radiotherapy.

All subsequent criteria apply only to the main study:

1. Subjects 21 years of age or older, with ECOG performance status 0 to 2, and life expectancy of more than 3 months with pathologically confirmed high grade astrocytic glioma, having undergone first-line standard-of-care therapy with surgery/biopsy followed by temozolomide and radiotherapy. Subjects with truncated adjuvant chemoradiotherapy may be enrolled at the Principal Investigator's discretion.

2. Documented tumor progression based on standard clinical, radiological or histological criteria, and deemed suitable for second line systemic therapy.

3. Sufficient tumor tissue available for PDO generation at baseline and at least one available or pending QPOP result.

4. Adequate organ function as defined by:

1. Bone marrow function

   1. Hemoglobin $\geq$ 9g/dl

   2. Absolute neutrophil count (ANC) $\geq$ 1.5 x 109/L

   3. Platelet count $\geq$ 100 x 109/L.

2. Liver function

   1. Bilirubin < 2.5x upper limit of normal (ULN)

   2. Alanine transaminase (ALT) and aspartate transaminase (AST) < 2.5x ULN or < 5x ULN if liver metastases are present

   3. Prothrombin time (PT) within the normal range for the institution.

3. Renal function

   1. Plasma creatinine <1.5x institutional ULN

5. Capable of swallowing tablets.

6. Recovery from any previous drug- or procedure-related toxicity to National Cancer Institute Common Terminology Criteria for Adverse Events (NCI-CTCAE) version 5.0 Grade 0 or 1 (except alopecia), or to baseline preceding the prior treatment.

**Exclusion criteria.**   All exclusion criteria apply to both the pre-screening phase and main study:

1. Chemotherapy, radiotherapy, surgery, immunotherapy or other therapy within 2 weeks of study entry.

2. Pregnancy or breastfeeding at the point where systemic anti-cancer therapy is initiated. Women of childbearing potential must have a negative pregnancy test at the point where systemic anti-cancer therapy is initiated. Women of childbearing potential and men, must agree to use adequate contraception (barrier method of birth control) while on anti-cancer treatment and until at least 3 months after the last study drug administration.

3. Concurrent cancer which is distinct in primary site or histology from the cancer being evaluated in this study EXCEPT cervical carcinoma in situ, treated basal cell carcinoma, superficial bladder tumors (Ta, Tis & T1) or any cancer curatively treated less than 5 years prior to study entry.

4. Subjects with leptomeningeal dissemination of disease and/or pure spinal high grade gliomas will be excluded.

5. Kidney disease which would clinically disqualify the subject from serial MRI scans with gadolinium contrast.

**Withdrawal criteria.**   Subjects may withdraw from the study and discontinue treatment for the following reasons, but are not limited to:

1. Disease progression,

2. Intercurrent illness that prevents further administration of treatment,

3. Unacceptable adverse event(s),

4. Subject becomes pregnant.

5. Subject decides to withdraw from the study, or

6. General or specific changes in the subject's condition render the subject unacceptable for further treatment in the judgment of the investigator.

**Subject replacement.**   Subjects will be replaced accordingly if:

1. Their QPOP analysis is unsuccessful, or its results do not lead to therapeutic decision making

2. The subject who withdraws or discontinues from the study prior to the completion of cycle 1 due to reasons other than toxicity or progressive disease.

   These subjects will not be included in our study endpoint analysis.

## Study assessments

A comprehensive list of clinical parameters will be collected and assessed as detailed in Table 1.

Briefly, safety assessments comprising of both clinical and laboratory assessments will be performed with two weeks of study enrolment, and at the start of each cycle of treatment. Clinical assessments include documenting the subjects' medical history, physical examination, evaluation of subjects' performance status (ECOG), and adverse event and toxicity assessments in accordance with the NCI-CTCAE version 5.0. Laboratory parameters include hematology (full blood count (FBC)), blood chemistries, and others as clinically indicated.

Radiologic assessments (standard MRI brain +/- perfusion, Ga68-NEB PET/MRI and DCE-MRI) will be performed within a week of any planned radiotherapy simulation of

**Table 1.  Overview of study assessments of the trial.**

| Assessment/Procedures | Within 2 weeks of study enrolment | At the start of each treatment cycle thereafter | 8 weeks after QPOP-guided systemic therapy | Final study visit |
|---|---|---|---|---|
| **Safety assessments** | | | | |
| *Clinical assessments* | | | | |
| Medical history | X | X | ■ | X |
| Physical examination | X | X | ■ | X |
| Concomitant medication notation | X | X | ■ | X |
| Evaluation of performance status (ECOG) | X | X | ■ | X |
| Adverse events and toxicity assessments (NCI-CTCAE version 5.0) | ■ | X | ■ | X |
| *Laboratory assessments* | | | | |
| Hematology (FBC) | X | X | ■ | ■ |
| Blood chemistries [a] | X | X | ■ | ■ |
| Others as clinically indicated [b] | X | X | ■ | ■ |
| **Radiologic assessments** | | | | |
| Standard MRI brain +/- perfusion | X | X | X | ■ |
| Ga68-NEB PET/MRI | X | X | X | ■ |
| DCE-MRI | X | X | X | ■ |

[a] Includes bilirubin, ALT, AST, ALP, creatinine, calcium, sodium, potassium

[b] Includes, but not limited to, serum pregnancy test only for female participants or women of childbearing potential

subjects during the pre-screening phase, as well as prior to and 8 weeks after exposure to QPOP-guided systemic therapies. Imaging on all other occasions will be routine and decided upon by the treating physician.

A final study visit comprising of a clinical assessment will be performed within 30 days from the time of study discontinuation or completion of study treatment.

## Ancillary studies

The protocol includes ancillary studies on machine-learning guided radiomics and tumor histopathology, and will be carried out for subjects who have given specific consent for these studies.

**Radiological study.** Standard MRI brain with or without perfusion sequences are typically performed at baseline and every 8 weeks until unequivocal progression in order to evaluate radiological response. In this study, we plan to add Ga68-NEB (Gallium-68 NOTA-Evans Blue) PET/MRI imaging and DCE-MRI imaging to examine BBB permeability and assess in-vivo drug effects on tumor vasculature over time. These studies are planned at 3 timepoints: 1) During the pre-screening phase, following confirmation of histological diagnosis and just prior to RT simulation and 2) during the main study, prior to and 8 weeks (+/- 1 week) after QPOP-guided systemic therapy. Radiological biomarkers will be correlated with radiological response based on follow up with routine MRI and clinical outcomes as well as therapeutic sensitivities.

**Radiomics study.** MRIs obtained at initial diagnosis and subsequent scans will be subjected to radiomic feature extraction, and correlated with spatially annotated histopathological biopsies opportunistically obtained as part of standard-of-care surgical resection both at initial diagnosis (chemotherapy- and radiotherapy-naïve) and at recurrence. Annotation of biopsy site would be done on T1-stereotaxy and pathology images would be correlated with spatial radiomic features and clustering. In addition, part of the biopsy will be used to generate high grade astrocytic glioma organoids, and tested for chemosensitivity towards both temozolomide and the optimal drug combination the participant will be treated with. These spatial properties of the high grade astrocytic glioma biopsies will be used for model building for prediction of post-chemotherapy recurrence.

**Histopathology study.** Paraffin-embedded blocks of high grade astrocytic glioma tumor will be stained to assess DNA damage response (γH2AX, RAD51), growth factor receptors (EGFR, PDGFR) and hypoxia markers (HIF1A) in relation to blood vessels and axons. Molecular analyses of TP53, ATRX, IDH1, IDH2, BRAF, and EGFRvIII mutations, MGMT methylation status, and 1p19q co-deletion would be performed as part of routine care.

**Plasma biomarkers study.** Ten milliliters of blood at baseline and prior to cycle 2 of the main study will be collected together with routine blood tests for plasma biomarkers analyses including circulating tumor DNA, proteomics and DNA methylation. These will be stored in the NUH tissue repository for future analysis.

## Ethics approval and consent to participate

This study has received ethical approval from National Healthcare Group (NHG) Domain Specific Review Board (DSRB), reference 2022/00103. All patients/ subject participants will give their written informed consent prior to any commencement of study-related assessments. The study methodologies conformed to the standards set by the International Conference on Harmonization of Technical Requirements for Registration of Pharmaceuticals for Human Use (ICH)–Good Clinical Practice (GCP) and Declaration of Helsinki.

## Discussion

HGG remains a deadly disease despite multimodality treatments including neurosurgery, radiotherapy and chemotherapy. Recurrence in HGG is almost universal and currently, there is no standard treatment for recurrent HGG. Clearly, there is an urgent need to develop better therapeutic strategies to improve outcomes of HGG patients.

Conventional trials are usually designed to study the efficacy of single agent or specific drug combination in a randomized fashion. Our study is unique because it is a personalized medicine study where patient-derived specimens are being assessed individually in response to an intervention without prior molecular profiling of the individual. Precision cancer medicine have been gaining traction in the field of neuro-oncology, with several clinical trials having demonstrated that molecular profiling of brain tumors is able to identify actionable mutations and guide cancer therapy for these patients [15, 37, 38]. However, the prognosis for HGG is still poor, with significant recurrence rates and few FDA-approved therapeutic agents for HGG. This is largely attributed to the well-reported intra-tumoral heterogeneity presented in tumors due to the presence of highly variable cancer cell subpopulations which limit the long-term efficacy of monotherapies [3, 4, 39]. Furthermore, evidence have demonstrated that monotherapies can drive clonal evolution through the development of *de novo* mutations in subclones, or the selection of treatment-resistant subclones, further limiting the promise the precision cancer medicine [40]. There is therefore a need to develop a combination therapy design which is able to target the diverse nature of HGG tumors. Moreover, biologic plasticity of cancer cells in response to systemic treatment is known to lead to altered therapeutic sensitivities over time [41]. A unique feature of our study is the generation of a matched pair of HGG organoids with TMZ/RT-resistance, simulating resistance induced by standard-of-care first-line treatment [26]. We posit that this may overcome the need for a second tissue harvest at the time of relapse as only a minority of HGG are eligible for surgery at the time of recurrence.

Importantly, our study leverages on the phenotypic response of patient-derived cancer organoids to determine the ideal recommended personalized combination therapy. In doing so, we can mitigate the challenges posed by the inherent heterogeneity presented in HGG tumors as combination treatment regimens are recommended purely based on the sensitivity of the tumor cells present in PDOs to the anti-cancer agents. While the culture conditions would not be able to support the vast heterogeneous landscape of HGG tumors, the conditions have been established to minimally retain the cancer stem cell population in HGG tumors, the main cancer cell population responsible for tumor resistance and recurrence [42, 43]. By utilizing HGG PDOs as an *ex vivo* model to elucidate the sensitivity of patient tumors to anti-cancer agents, we can therefore identify effective therapeutic strategies that targets the HGG cancer stem cells that are individualized to each patient. Indeed, two clinical trials in Sweden and Norway have commenced trials in which sensitivities of individualized patient cancer stem cells to FDA-approved drugs are used recommend novel therapeutic strategies for patients with recurrent glioblastoma (NCT05380349, NCT05043701). The evidence therefore supports the rationale underlying our study of using HGG PDOs as a platform to identify personalized treatment alternatives for patients who have progressed on from the standard-of-care TMZ/RT, as well as the increasingly need for individualized therapy for patients with HGG.

## Supporting information

**S1 File. SPIRIT 2013 checklist.** Recommended items to address in a clinical trial protocol and related documents.
(PDF)

**S2 File. Original study protocol.**
(PDF)

## Acknowledgments

We thank all investigators who devote their time and passion in the implementation of the study and the recruitment of subjects.

## Author Contributions

**Conceptualization:** Tan Boon Toh, Balamurugan A. Vellayappan, Edward Kai-Hua Chow, Dean Ho, Tseng Tsai Yeo, Andrea Li Ann Wong.

**Funding acquisition:** Tan Boon Toh, Dean Ho, Tseng Tsai Yeo, Andrea Li Ann Wong.

**Investigation:** Tan Boon Toh, Nagarjun Bolem, Balamurugan A. Vellayappan, Bryce Wei Quan Tan, Yating Shen, Yvonne Li En Ang, Nivedh Dinesh, Kejia Teo, Vincent Diong Weng Nga, Shiong Wen Low, Pek Lan Khong, Edward Kai-Hua Chow, Dean Ho, Tseng Tsai Yeo, Andrea Li Ann Wong.

**Methodology:** Tan Boon Toh, Andrea Li Ann Wong.

**Project administration:** Sou Yen Soon.

**Resources:** Tan Boon Toh, Andrea Li Ann Wong.

**Software:** Edward Kai-Hua Chow, Dean Ho.

**Supervision:** Tan Boon Toh, Andrea Li Ann Wong.

**Writing – original draft:** Tan Boon Toh, Nagarjun Bolem, Bryce Wei Quan Tan, Sou Yen Soon, Pek Lan Khong, Andrea Li Ann Wong.

**Writing – review & editing:** Tan Boon Toh, Dexter Kai Hao Thng, Andrea Li Ann Wong.

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
