## [Decision Letter · Decision Letter 0]

7 Jun 2024

PONE-D-23-34301Evaluation of ex vivo drug combination optimization platform in recurrent high grade astrocytic glioma: An interventional, non-randomized, open-label trial protocol

PLOS ONE

Dear Dr. Toh,

Thank you for submitting your manuscript to PLOS ONE. After careful consideration, we feel that it has merit but does not fully meet PLOS ONE’s publication criteria as it currently stands. Therefore, we invite you to submit a revised version of the manuscript that addresses the points raised during the review process.

We look forward to receiving your revised manuscript.

Kind regards,

Kevin Camphausen

Academic Editor

PLOS ONE

 [ALAW received funding from the National University Health System (NUHS) Seed Fund (NUHSRO/2021/052/RO5+6/Seed-Mar/01) and TTB obtained funding from the National University of Singapore Institute for Digital Medicine (WisDM) Program Seed Fund (WisDM/Seed/002/2021).].  

[I have read the journal's policy and the authors of this manuscript have the following competing interests: EKHC and DH are shareholders in KYAN Technologies.]. 

Reviewers' comments:

Reviewer's Responses to Questions

**Comments to the Author**

1. Does the manuscript provide a valid rationale for the proposed study, with clearly identified and justified research questions?

Reviewer #1: Partly

Reviewer #2: Yes

2. Is the protocol technically sound and planned in a manner that will lead to a meaningful outcome and allow testing the stated hypotheses?

Reviewer #1: Partly

Reviewer #2: Partly

3. Is the methodology feasible and described in sufficient detail to allow the work to be replicable?

Reviewer #1: Yes

Reviewer #2: No

4. Have the authors described where all data underlying the findings will be made available when the study is complete?

Reviewer #1: Yes

Reviewer #2: Yes

5. Is the manuscript presented in an intelligible fashion and written in standard English?

Reviewer #1: Yes

Reviewer #2: Yes

6. Review Comments to the Author

You may also provide optional suggestions and comments to authors that they might find helpful in planning their study.

Reviewer #1: This is a thought provoking outline of a protocol to evaluate drug combinations in recurrent HGG, using ex vivo methods. It addresses a significant question that could provide much needed information on improving treatment of those with recurrent HGGs. Nevertheless greater clarification is needed in the proposed protocol before acceptance, particularly in terms of study justification and evaluation of outcomes. However, with additions the protocol will be suitable for publication.

1) Further information of PDOs and QPOP techniques, particularly in how PDOs differ from 2D cell cultures would be helpful. Additionally, specific data supporting claims of PDOs faithfully recapitulate tumor heterogeneity, correlate with mutational status, and are an effective tool for determining drug response should be presented in the Intro text.

2)Line 106: "Current biomarker based approaches in predicting drug sensitivity and clinical outcome are limited only in select patient subpopulations and specific drugs and are still not broadly applicable". Should this read as "outcomes are limited only TO patient subpopulations?

3)Line 173: Is the criteria to determine treatment feasibility limited to "if there is successful organoid generation and QPOP analysis in >50% of samples collected."? Is this of all tumor samples regardless of QPOP combination (QPOP-1, QPOP-2 and QPOP-3) or is it limited to QPOP-1? Is there any additional criteria? What about patients that have two of the three combinations? Greater clarification is needed.

4)Line 198: Is there a set schedule for TMZ exposure and ionizing radiation in treatment-naive organoids?

5)Line 247: What criteria or information will be used to determine if a patient will receive category 2 combination or monotherapy?

6) Line 295: What is the justification for excluding young adults (ages 18-21) from the study?

7) Line 322: Will patients with original low grade gliomas who's undergo transformation to HGG be considered?

8) Line 323 First-line standard therapy should be clarified as the stupp protocol. Are therapies such as TTFs allowed?

9) Line 329: What determines an adequate PDO generation? Is it simply any successful QPOP analysis?

10) Line 381: How will subject replacement impact assessment of feasibility?

11) Line 718: How will tumor from any subsequent procedure be used if not QPOP-1 or QPOP-3?

12) Figure 1: Is there really no information on history/physical at pre-screening that would assess patient appropriateness?

Reviewer #2: This paper presented a protocol of a pilot study for feasibility with N=10. The design itself is straightforward. The major weakness is that there is no statistical consideration including statistical data analysis plan presented.

Besides that, the following weakness is noted:

1. Three specific aims are presented but based on the current wording, some aims can not be achieved. For example, Aim 2 is to determine the utility of QPOP-derived drug combination in treating recurrent HGG. With the limited sample size, this aim is unlikely to be completed. Therefore, refining the aims is recommended.

2. Two efficacy endpoints are six-month progression-free survival (PFS6) and one-year overall survival(OS12). It is not appropriate to say that PFS6 and OS12 refers to the percentage of subjects who are alive at 6 month and free of HGG progression and alive at 12 months, respectively.

7. PLOS authors have the option to publish the peer review history of their article (what does this mean?). If published, this will include your full peer review and any attached files.

Reviewer #1: No

Reviewer #2: No

---

## [Author Response · Author response to Decision Letter 0]

8 Jul 2024

Response: We have now formatted our manuscript according to PLOS ONE’s style requirements.

2. Thank you for stating the following financial disclosure: [ALAW received funding from the National University Health System (NUHS) Seed Fund (NUHSRO/2021/052/RO5+6/ Seed-Mar/01) and TTB obtained funding from the National University of Singapore Institute for Digital Medicine (WisDM) Program Seed Fund (WisDM/Seed/002/2021).].

Response: We acknowledge that the funders, NUHS and WisDM, had no role in study design, data collection and analysis, decision to publish, or preparation of the manuscript. We have thus included the statement in the cover letter and in the manuscript on page 27 (lines 587-589).

[I have read the journal's policy and the authors of this manuscript have the following competing interests: EKHC and DH are shareholders in KYAN Technologies.]. 

Please confirm that this does not alter your adherence to all PLOS ONE policies on sharing data and materials, by including the following statement: "This does not alter our adherence to PLOS ONE policies on sharing data and materials.” If there are restrictions on sharing of data and/or materials, please state these. Please note that we cannot proceed with consideration of your article until this information has been declared. 

Response: We acknowledge that there are restrictions on the provision of the code for QPOP analysis as it is proprietary to Kyan Technologies. We have thus updated this in the cover letter and in the Data Availability statement in the manuscript on page 27 (lines 579-580).

4. We note that you have indicated that there are restrictions to data sharing for this study. PLOS only allows data to be available upon request if there are legal or ethical restrictions on sharing data publicly. 

b) If there are no restrictions, please upload the minimal anonymized data set necessary to replicate your study findings to a stable, public repository and provide us with the relevant URLs, DOIs, or accession numbers. You also have the option of uploading the data as Supporting Information files, but we would recommend depositing data directly to a data repository if possible.

Response: While no datasets were generated from this manuscript, We acknowledge that subsequent data culminating from this study will be published in international journals and deidentified research data from the study will be made publicly available upon request once the study is complete. We have now updated the Data Availability statement on page 27 (lines 577-578).

Response: We have now reorganised the manuscript and included the ethics statement in the Methods/design section of the manuscript on page 23 (lines 488-495).

6. Please include captions for your Supporting Information files at the end of your manuscript, and update any in-text citations to match accordingly. 

Response: We have now included captions for the SPIRIT checklist uploaded as a Supporting Information file at the end of the manuscript on page 32 (lines 799-801).

Review Comments to the Author

Reviewer #1: This is a thought provoking outline of a protocol to evaluate drug combinations in recurrent HGG, using ex vivo methods. It addresses a significant question that could provide much needed information on improving treatment of those with recurrent HGGs. Nevertheless greater clarification is needed in the proposed protocol before acceptance, particularly in terms of study justification and evaluation of outcomes. However, with additions the protocol will be suitable for publication.

1) Further information of PDOs and QPOP techniques, particularly in how PDOs differ from 2D cell cultures would be helpful. Additionally, specific data supporting claims of PDOs faithfully recapitulate tumor heterogeneity, correlate with mutational status, and are an effective tool for determining drug response should be presented in the Intro text.

Response: We are grateful to the reviewer for his/her helpful suggestions. Regarding the distinctions between PDOs and 2D cell cultures, organoids are stable 3-dimensional systems which can self-organise and replicate the multicellularity, architecture and functionality similar to those of the actual organs.1, 2 In the context of HGG specifically, organoids derived from patient-derived material faithfully recapitulated the heterogeneous cell identity observed in parental tumours.3, 4 This was evidenced in the maintenance of the heterogeneous expression of stem cell markers such as SOX2, OLIG2 and NESTIN.3, 4 Furthermore, the HGG PDOs were shown to retain the mutational profiles of parental tumours, including genetic aberrations frequently observed in gliomas such as PTEN loss and the gain-of-function EGFR variant III (EGFRvIII).3 Correspondingly, drug sensitivity studies in HGG PDOs successfully correlated the molecular profiles of PDOS to the drug sensitivity and resistance profiles of PDOs. For instance, HGG PDOs harbouring EGFR aberrations exhibited preferential sensitivity to first-generation EGFR inhibitor, gefitinib, while the presence of secondary mutations rendered the PDOs resistant.3 Human PDOs therefore represent more clinically relevant models compared to traditional cell line-based models, with the ability to recapitulate disease heterogeneity for drug response studies with significant clinical relevance.

As an experimental-analytical hybrid drug combination interrogation platform, QPOP can model the expected patient response to panel of drugs by systematically perturbing biological model systems such as patient-derived cells or organoids to the drugs in defined combinations derived from an orthogonal-array composite design.5 The functional responses of the patient-derived avatars are then used to establish a second-order regression model from which all possible combinations of the drug panel can be ranked.5 QPOP has been successfully applied in patients with refractory hematological malignancies such as multiple myeloma and lymphomas, yielding favourable clinical response in patients.5, 6

These additional information and description are now included in the Introduction of the manuscript on pages 5-6 (lines 92-98, 103-106 and 124-132).

2)Line 106: "Current biomarker based approaches in predicting drug sensitivity and clinical outcome are limited only in select patient subpopulations and specific drugs and are still not broadly applicable". Should this read as "outcomes are limited only TO patient subpopulations?

Response: We sincerely apologise for the typographical error, and have now rectified the statement to read “outcomes are limited to select patient subpopulations” in the manuscript on page 6 (line 115).

3)Line 173: Is the criteria to determine treatment feasibility limited to "if there is successful organoid generation and QPOP analysis in >50% of samples collected."? Is this of all tumor samples regardless of QPOP combination (QPOP-1, QPOP-2 and QPOP-3) or is it limited to QPOP-1? Is there any additional criteria? What about patients that have two of the three combinations? Greater clarification is needed.

Response: We apologise for the lack of clarity and thank the reviewer for highlighting this important point. The criteria to determine feasibility refers to the generation of a QPOP combination in >50% of all tumour samples. We expect all our patients to have a QPOP-1 sample (i.e. first resection) but not everyone will have QPOP-2 and QPOP-3 samples for organoid generation as only approximately 25% of patients who recur would be eligible for repeat surgery.7 The criteria for successful QPOP analysis has now been clarified in the manuscript on page 8 (lines 185-186).

4)Line 198: Is there a set schedule for TMZ exposure and ionizing radiation in treatment-naive organoids?

Response: We sincerely thank the reviewer for his/her valuable feedback. Established HGGs were subjected to artificial delivery of temozolomide (TMZ) and radiotherapy (RT) to simulate the Stupp Protocol in vitro as adapted from protocols established previously.8 Briefly, HGG PDOs were subjected to 2 Gy of irradiation every alternate day over a period of 10 days. Additionally, PDOs were treated with temozolomide at the maximum serum concentration of 37.6µM to mimic physiological conditions prior to radiation exposure.9 The set schedule for temozolomide and ionizing radiation exposure is not included in the manuscript as it is currently being optimised and validated.

5)Line 247: What criteria or information will be used to determine if a patient will receive category 2 combination or monotherapy?

Response: We thank the reviewer for the question. We would like to clarify that patients will receive combination therapy over monotherapy unless QPOP combination therapy results do not fall within the criteria for usage as category 1 or category 2 combinations. A category 1 recommendation is where the combination generated has published data in the setting of gliomas, whereas a category 2 recommendation is where the combination generated has published data only in non-glioma settings, but where the individual agents have shown CNS/ anti-glioma activity. These scenarios would be prioritised over monotherapy and monotherapy will only be applied when QPOP combination therapy results do not fall within the criteria for usage. We have since clarified this in the manuscript on pages 12-13 (lines 280-283 and 288).

6) Line 295: What is the justification for excluding young adults (ages 18-21) from the study?

Response: We appreciate the reviewer for his/her question. Young adults from the ages of 18-21 were excluded for the study due to the legality of trial conduct as stipulated by the Health Science Authorities of Singapore, the presiding authorities over trial conduct in Singapore. Individuals under the age of 21 are deemed to be minors and hence would not be able to provide informed consent unless legally represented. Young adults aged 18-21 were thus excluded as inclusion would require a different trial conduct.

7) Line 322: Will patients with original low grade gliomas who's undergo transformation to HGG be considered?

Response: We sincerely thank the reviewer for the important question. Only patients with a first suspected then histologically-confirmed diagnosis of a high grade astrocytic glioma may be enrolled, hence those with low grade gliomas will be excluded. However, should the low grade glioma subsequently undergo transformation to be radiologically suspicious and histologically-confirmed for HGG, the subject may be enrolled onto the study at that juncture, provided he/ she meets other eligibility criteria and has availability of tissue resection post-surgery.

8) Line 323 First-line standard therapy should be clarified as the stupp protocol. Are therapies such as TTFs allowed?

Response: We are grateful to the reviewer for his/her feedback. As this is a trial of second-line therapy, first-line therapy is mainly relevant in the context of study eligibility. The protocol specifically requires patients to have received standard-of-care therapy with adjuvant temozolomide and radiotherapy; TTF is not required and hence patients who have had TTF therapy may be enrolled, but TTF is not mandated as it is not regarded as standard-of-care therapy. 

9) Line 329: What determines an adequate PDO generation? Is it simply any successful QPOP analysis?

Response: We thank the reviewer for the valuable question. For the study protocol, adequate PDO generation refers to the successful establishment for PDOs which can be subjected to downstream QPOP analysis. During the course of the QPOP analysis, measures such as the z’-score of the assay and model diagnostics would be evaluated to assess the robustness of the assay. Only adequate PDO establishment would yield a satisfactory z’-score (z’ > 0.5) and model diagnostics (R2 > 0.7) that gives confidence in the robustness of the analysis. Hence, a successful QPOP analysis is thus contingent on adequate PDO generation and is a surrogate measure for PDO establishment. The criteria for successful QPOP analysis is now defined in the manuscript on pages 8-9 (lines 186-188).

10) Line 381: How will subject replacement impact assessment of feasibility?

Response: We are grateful to the reviewer for his/her feedback. In the trial protocol, subjects will be replaced if the QPOP analysis was unsuccessful or if the subject withdraws due to reasons other than toxicity or progressive disease. In both scenarios, the replacement of subjects ensures that the assessment of feasibility will not be understated. In the former scenario, the quality of the QPOP assay would be determined through assessment of the z’-score and model diagnostics as mentioned in response to question 9. Hence, subjects should be replaced as results from a QPOP analysis which failed the quality assessment would not be provided to the clinicians for patient treatment. Consequently, the case would not contribute to the assessment of feasibility of QPOP as an ex vivo drug combination optimization platform. Furthermore, the quality of the QPOP assay is contingent on the generation of HGG PDOs and does not reflect the feasibility of using QPOP. Similarly, the latter situation would result in an understatement in the evaluation of QPOP as a feasible platform for drug combination optimisation. Hence, patient replacement was implemented to minimise the underrepresentation of the evaluation results.

11) Line 718: How will tumor from any subsequent procedure be used if not QPOP-1 or QPOP-3?

Response: We appreciate the reviewer for this question. Tumour not utilised for the generation of PDOs for QPOP analysis would be used for ancillary histopathology studies to provide additional characterisation of patient tumours to guide the treatment-decision making process as indicated on page 22 (lines 474-480). 

12) Figure 1: Is there really no information on history/physical at pre-screening that would assess patient appropriateness?

Response: We thank the reviewer for his/her query. We did not include assessment of medical history and physical examination as at pre-screening as all patients enrolled onto the study will need to be fit for general anaesthesia and adjuvant chemoradiotherapy, implying a minimum level of fitness. As the therapeutic element of this study commences only upon enrolment onto the main study, medical history and physical

---

## [Editor Report · Decision Letter 1]

12 Jul 2024

Evaluation of ex vivo drug combination optimization platform in recurrent high grade astrocytic glioma: An interventional, non-randomized, open-label trial protocol

PONE-D-23-34301R1

Dear Dr. Toh,

We’re pleased to inform you that your manuscript has been judged scientifically suitable for publication and will be formally accepted for publication once it meets all outstanding technical requirements.

Kind regards,

Kevin Camphausen

Academic Editor

PLOS ONE
---

## [Editor Report · Acceptance letter]

17 Jul 2024

PONE-D-23-34301R1 

PLOS ONE

Dear Dr. Toh, 

I'm pleased to inform you that your manuscript has been deemed suitable for publication in PLOS ONE. Congratulations! Your manuscript is now being handed over to our production team.

Kind regards, 

on behalf of

Dr. Kevin Camphausen 

Academic Editor

PLOS ONE